# Flexible Docking via Unbalanced Flow Matching

**Gabriele Corso** [* 1]  **Vignesh Ram Somnath** [* 2]  **Noah Getz** [1]
**Regina Barzilay** [1]  **Tommi Jaakkola** [1]  **Andreas Krause** [2]

## Abstract

Diffusion models have emerged as a recent successful paradigm for molecular docking. However, these methods treat the protein either as a rigid structure, or force the model to fold proteins from unstructured noise. In this work, we instead focus on flexible docking, leveraging the unbound distribution of proteins to model the precise effect(s) of ligand binding. While Flow Matching (FM) presents an attractive option for this task, we show that a naive application of flow matching results in a complex learning task with poor performance. We thus propose *Unbalanced Flow Matching*, a generalization of flow matching that allows us to tradeoff sample efficiency with approximation accuracy by relaxing the marginal constraints. Empirically, we validate our framework on flexible docking, demonstrating strong improvements in protein conformation prediction while retaining comparable docking accuracy.

## 1. Introduction

Molecular docking predicts the binding structure between proteins and small molecules, a crucial interaction for the mechanism of action of most drugs. Over the past decades, significant progress has been made in molecular docking, initially through classical search techniques (Alhossary et al., 2015; McNutt et al., 2021) and more recently with DL-based regression (Stärk et al., 2022) and diffusion models (Corso et al., 2022). However, these methods primarily focus on rigid docking, assuming the protein has a fixed structure. While this assumption is realistic in some scenarios, it severely limits the applicability of these methods.

Existing flexible docking methods have so far failed to provide satisfactory levels of accuracy. Traditional search-

---

[*]Equal contribution  [1]CSAIL, MIT  [2]Department of Computer Science, ETH Zurich. Correspondence to: Gabriele Corso <gcorso@csail.mit.edu>, Vignesh Ram Somnath <vsomnath@inf.ethz.ch>.

*Accepted at the 1st Machine Learning for Life and Material Sciences Workshop at ICML 2024.* Copyright 2024 by the author(s).

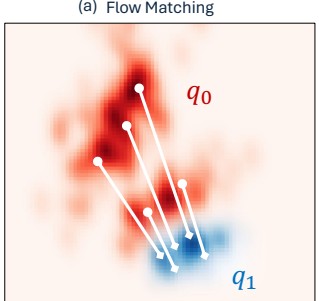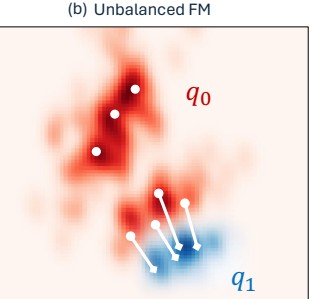

*Figure 1.* Comparison between the mappings learnt with Flow Matching (left) and Unbalanced Flow Matching (right).

based methods struggle to account for protein degrees of freedom due to the significantly increased dimensionality of the search space. Deep learning methods have improved on this by extending diffusion processes to include the protein (Qiao et al., 2024; Abramson et al., 2024), but often force the model to fold proteins from unstructured noise (cofolding), resulting in structure predictions that are frequently worse than the inputs. To avoid this issue, it is necessary to directly map the distribution of unbound protein structures to those of structures bound to a given ligand.

Flow matching is a recent, generative modelling framework, capable of learning a transport between arbitrary distributions. However, its direct application to this problem, where the two distributions are highly structured, results in a complex learning task, and poor performance. To overcome these challenges, we propose Unbalanced Flow Matching, a new framework for learning a transport between distributions where we relax the marginal constraints of FM and study a larger class of (partial) maps between the two distributions (intuitive illustration in Figure 1). We demonstrate theoretically how trading off some sample efficiency, Unbalanced FM allows one to define significantly simpler maps resulting in improved performance.

Empirically, we demonstrate that our new modeling perspective, enhances structure prediction quality, especially for protein conformations. On the PDBBind benchmark, our approach FLEXDOCK improves the proportion of very accurate protein structure predictions (all-atom RMSD $< 1$Å) from 39.8% to 44.1%, while retaining comparable docking

accuracy (ligand RMSD < 2Å). On the PoseBusters benchmark dataset, FLEXDOCK outperforms most co-folding methods, despite only being trained on the PDBBind dataset.

## 2. Background and Related Work

**Flow matching (FM)** (Lipman et al., 2022; Albergo et al., 2023) is a generative modeling paradigm that was introduced as a flexible generalization of diffusion models, and allows learning a transport between arbitrary distributions with a simulation-free objective. Given two distributions $q_0$ and $q_1$, FM provides a way of learning a vector field $v_t$ which induces a continuous normalizing flow $\psi_t(x)$ that transports $q_0$ to $q_1$, i.e., $q_1(x) = [\psi_1]_\# q_0(x))$, where $\#$ denotes the pushforward operator.

The key idea in FM is defining a conditional flow $\psi_t(x_0|x_1)$ interpolating between $x_0 \sim q_0$ and $x_1 \sim q_1$, and its associated vector field $u_t(x_t|x_1) = \frac{d}{dt}\psi_t(x_t|x_1)$. One can then learn the marginal vector field $\hat{v}_t(x, t; \theta)$ with a neural network ($\hat{v}_t(x, t; \theta) \approx v_t(x, t)$, by regressing against the conditional vector field with the CFM objective:

$$\mathcal{L}_{\text{CFM}} = \mathbb{E}_{t, x_0 \sim q_0, x_1 \sim q_1} \|v_t(x_t; \theta) - u_t(x_t|x_1)\|^2 \quad (1)$$

FM was further generalized by Pooladian et al. (2023) and Tong et al. (2023) which showed that the sampling distribution in the CFM objective, which we will refer to as coupling distribution, does not have to be independent samples from $q_0$ and $q_1$ and can be an arbitrary joint distribution $q(x_0, x_1)$ as long as it satisfies the marginal constraints being $q_0$ and $q_1$ respectively. This formulation enabled drawing a connection between FM and optimal transport (OT). When using OT to define the coupling distribution $q$, the flows become straight and the transport cost $\mathbb{E}_{q_0(x_0)} \|\psi_1(x_0) - x_0\|^2$ is the OT cost $W_2^2(q_0, q_1)$.

**Protein-ligand binding** When proteins bind to small molecules, their structural distribution adjusts to fit the molecule. Understanding this conformational change is critical for accurately predicting binding interactions, and computational methods for this fall into two categories: *co-folding* and *flexible docking*.

*Co-folding* involves predicting the bound structure of the protein and the ligand from scratch as a single task. Based on the success of AlphaFold 2 (AF2) (Jumper et al., 2021) for protein structure prediction, a number of methods have extended AF2 for small-molecule co-folding (Qiao et al., 2024; Bryant et al., 2023; Krishna et al., 2024; Abramson et al., 2024). While these have achieved varied success, they typically require large amounts of training data and have slow inference times. (Wang et al., 2023) adopts a *co-folding* strategy based on diffusion processes, but instead of the Euclidean space, the diffusion processes are defined on

the backbone torsion angles of the protein, and the product space of rotations, translations and torsions for the ligand.

**Protein Conformational Changes and Flexible Docking**
Flexible docking assumes access to unbound structures of proteins (known as apo) and predicts how these will change upon ligand binding (producing holo-structures). Since the conformational change is usually small and localized due to the molecule's size and energetic impact, this approach has been preferred for protein-ligand structure prediction, making it suitable for large-scale screening pipelines.

Traditional *search-based* docking methods define a scoring function and search the space of possible poses (through rigid movement and torsion angle changes of the ligand) to find the minimum of the scoring function (Alhossary et al., 2015; McNutt et al., 2021). These methods can typically, incorporate protein flexibility by adding torsion angles of the sidechains in the pocket to the search space. However, due to the increased dimensionality and the protein's flexibility beyond the sidechains, traditional methods struggle to find optimal joint poses.

Recently, a number of deep learning methods have been proposed that leverage the flexibility of proteins in molecular docking. DIFFDOCK-POCKET (Plainer et al., 2023) and RE-DOCK (Huang et al., 2024) use diffusion and diffusion bridge models to model the flexibility in protein pocket sidechains in addition to ligand flexibility. DYNAMICBIND (Lu et al., 2024), the closest related work to ours, incorporates backbone flexibility, with hardcoded noise perturbation rules that interpolate from apo residue frames to holo residue frames. While the operate in the blind docking setting, they do not directly model the atomic positions, leaving limited utility for downstream applications such as binding affinity calculation. (Somnath et al., 2023) explicitly predicts the conformational changes between apo and holo states of proteins using Diffusion Schrödinger Bridges, by treating the apo and holo states as paired data. While they account for the constraint that minibatch-OT maps cannot be computed as in Pooladian et al. (2023), directly interpolating between conformational states suffer from the same complex learning task as we outlined in Section 3.2. Furthermore, their method was not evaluated on the flexible docking task.

## 3. Unbalanced Flow Matching

In this section, we first explain, using the specific example of the flexible docking task, the issue with existing approaches and the motivation for the development of a new technique for learning a transport between two distributions. Then, Section 3.2 introduces Unbalanced Flow Matching, Section 3.3 provides a theoretical formalization of the efficiency vs approximation tradeoff, and Section 3.4 discusses the choice of coupling distribution and its link to Unbalanced OT.

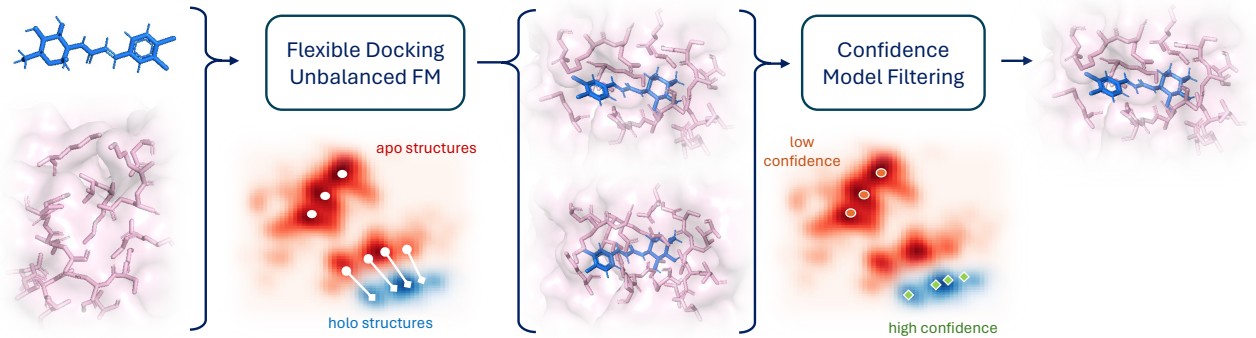

*Figure 2.* Overview of our flexible docking pipeline through Unbalanced Flow Matching.

### 3.1. Motivation and Overview

The main motivation for flexible docking over co-folding is to leverage the unbound distribution of proteins and focus exclusively on modeling the precise effects of ligand binding. Our goal, therefore, is to define the task in such a way that the model only needs to learn these small adjustments, rather than refolding the protein entirely. Diffusion modeling approaches for this task (Qiao et al., 2024; Abramson et al., 2024) force the model to largely refold the protein's structure because approximating the prior distribution from holo-structures (sampled during training) to that of apo-structures (sampled during inference) requires large noise levels.

Flow matching offers the compelling alternative of simply using the distribution of apo-structures as the initial $q_0$ and building a flow to the distribution of holo-structures $q_1$. However, this has two critical issues that arise when looking at the specific task. Firstly, because X-ray crystallography is the main source of bound conformations, for most complexes in our training data we have a single holo-structure. This prevents us from using minibatch-OT based flow matching techniques (Pooladian et al., 2023), leading to a large expected length of conditional flows.

Secondly, even if one bypasses the issue of single samples from the bound structures during training through expensive methods like NMR or extended molecular dynamics, the OT cost will likely remain very high. In fact, although the different conformational states of the protein do not change significantly upon ligand binding, their relative weights are often notably altered. Specifically, the protein typically reduces its entropy upon binding, as only a subset of the apo conformations allow for ligand binding (with minor induced fit). This common scenario results in a transport between apo and holo distributions that, even in the optimal setting (as represented in Figure 1.a), requires the model to move the protein between conformations, leading to large conditional and marginal flows. As discussed in Pooladian et al. (2023) and Benton et al. (2023), these large flows can result

in a complex learning task and significant approximation errors. In lieu of these issues, we develop the Unbalanced Flow Matching framework.

*Table 1.* **Flow Matching (FM) vs Unbalanced Flow Matching (FM)**. PDBBind docking performance with a small model (4M)

| Method | Ligand RMSD | |
|---|---|---|
| | % < 2Å↑ | % < 5Å↑ |
| FLEXDOCK (FM) (10) | 2.6 | 38.9 |
| FLEXDOCK (UFM) (10) | 10.6 | 53.1 |

### 3.2. Unbalanced Flow Matching

In the generalized Flow Matching formulation presented by Pooladian et al. (2023) the coupling distribution $q(\mathbf{x}_0, \mathbf{x}_1)$ is constrained to have the marginals of each variable being, respectively, $q_0$ and $q_1$. This condition is key to guarantee that the pushforward of $q_0$ *under the optimal flow* is a $q_1$ i.e. that we can sample I.I.D. $q_1$ by sampling $q_0$ and transporting the particle through the flow. However, for many structured distributions, this condition also causes the resulting mappings to be complex and have a high expected length (Figure 1.a). Benton et al. (2023) demonstrated how this complexity in learning the vector field of the flow translated into a mismatch between the true and learned distributions.

Unbalanced Flow Matching relaxes this constraint to obtain significantly shorter and simpler flows (See Table 1). By not imposing any hard constraints on the coupling $q$, we aim to keep the expected mapping cost between pairs $(\mathbf{x}_0, \mathbf{x}_1) \sim q$ low making the learning task easier. The objective function, in Euclidean space, remains:

$$\mathcal{L}_{\text{UFM}} = \mathbb{E}_{t,(\mathbf{x}_0, \mathbf{x}_1) \sim q} \|v_t(\mathbf{x}_t; \theta) - u_t(\mathbf{x}_t|\mathbf{x}_1)\|^2 \quad (2)$$

However, with arbitrary coupling distributions $q$, even if the vector field is learned perfectly its pushforward of $q_0$ will no longer correspond to $q_1$. To obtain unbiased samples from $q_1$ we will have to use techniques like rejection sampling to

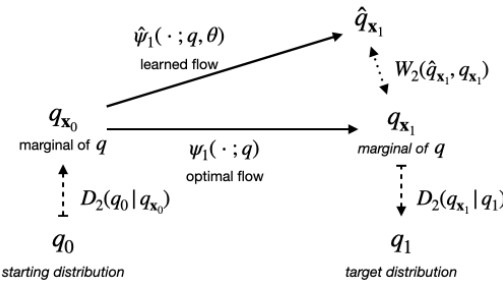

*Figure 3.* Relationship between the different distributions introduced in the theoretical analysis.

reweight the generated samples by their relative likelihood under $q_0$ and $q_1$ vs the marginals of $q$, which we indicate with $q_{\mathbf{x}_0}$ and $q_{\mathbf{x}_1}$. In the following sections, we formalize and analyze the tradeoff between sample efficiency and sample quality that arises when using Unbalanced FM and discuss how to choose a coupling distribution $q$ that obtains a balance on the optimal Pareto frontier.

### 3.3. Efficiency vs Approximation Tradeoff

Let $\psi_1(\cdot; q)$ be the optimal flow from the Unbalanced FM objective with couplings $q$ and $\hat{\psi}_1(\cdot; q, \theta)$ its approximation we are able to learn. FM guarantees us that $q_{\mathbf{x}_1} = [\psi_1(\cdot|q)]_{\#}q_{\mathbf{x}_0}$ and let $\hat{q}_{\mathbf{x}_1} = [\hat{\psi}_1(\cdot|q, \theta)]_{\#}q_{\mathbf{x}_0}$. A summary of all the defined distributions and their relationship is provided in Figure 3.

The definition of Unbalanced FM as a method to bridge two distributions $q_0$ and $q_1$ leads us to analyze the tradeoff between the approximation error when learning the flow, formalized as $W_2^2(q_{\mathbf{x}_1}, \hat{q}_{\mathbf{x}_1})$, and the sample efficiency $\text{ESS}^*(q)$ that derives from having to perform rejection sampling to bridge the gaps between $q_0$ and $q_{\mathbf{x}_0}$ and between $q_{\mathbf{x}_1}$ and $q_1$. Simple mappings will result in low approximation errors but potentially lower efficiency, and vice versa.

The tradeoff in the choice of optimal coupling $q^*$ can be expressed as a joint objective:

$$q^* = \min_{q} \; \alpha \, W_2^2(\hat{q}_{\mathbf{x}_1}, q_{\mathbf{x}_1}) - \beta \, \log \text{ESS}^*(q) \quad (3)$$

Maximizing sample efficiency, setting $\alpha << \beta$, recovers flow matching, while minimizing allowed approximation errors, setting $\alpha >> \beta$, translates into pure rejection sampling. Below we provide bounds for each of the two components that will lead us to better understand the class of optimal coupling distributions.

**Approximation error** Using Theorem 1 from Benton et al. (2023) we can show (proof in Appendix A.1) that the approximation error for a given coupling distribution $q$, $W_2^2(\hat{q}_{\mathbf{x}_1}, q_{\mathbf{x}_1})$, is bounded by the expected transport cost of the coupling $q$:

**Proposition 1.** *Under appropriate assumptions, we have:*

$$W_2^2(\hat{q}_{\mathbf{x}_1}, q_{\mathbf{x}_1}) \leq \mathbb{E}_{(\mathbf{x}_0, \mathbf{x}_1) \sim q} \|\mathbf{x}_0 - \mathbf{x}_1\|^2 \cdot L^2. \quad (4)$$

*where $L$ is the exponential of an integral over $t$ on the bound of the Lipschitz constant of the learned vector field.*

**Sample efficiency** We can measure the sample efficiency of the model when transporting samples from $q_0$ to unbiased samples of $q_1$ via the effective sample size $\text{ESS}^*(q)$, i.e. the reciprocal of how many samples from $q_0$ it takes using the ideal flow $\psi_1(\cdot, q)$ to generate an unbiased sample from $q_1$. In Proposition 2 (derivation in Appendix A.2), we demonstrate this sample efficiency is bounded by the similarity between the $q_0$ and $q_1$ and the respective marginals of $q$:

**Proposition 2.** *The effective sample size, $\text{ESS}^*$, for sampling $q_1$ when having access to samples of $q_0$ and a perfectly trained flow with coupling distribution $q$ is bounded by:*

$$\text{ESS}^*(q) \geq \exp\left[-D_2(q_0|q_{\mathbf{x}_0}) - D_2(q_{\mathbf{x}_1}|q_1)\right] \quad (5)$$

*where $D_2$ is the Rényi Divergence of order 2.*

### 3.4. Choosing the coupling

An obvious choice of couplings are those derived from Unbalanced Optimal Transport. Unbalanced OT relaxes the mass conservation constraint of optimal transport allowing to trade it off with reductions in mapping costs. In particular, the static unbalanced OT problem looks for coupling distributions $q$ that optimally balance the expected mapping cost and the preservation of the marginals via the objective (Séjourné et al., 2023):

$$\text{UOT}(q_0, q_1) \triangleq \min_{q} \mathbb{E}_{(\mathbf{x}_0, \mathbf{x}_1) \sim q}[C(\mathbf{x}_0, \mathbf{x}_1)] + \\ + D_{\varphi_0}(q_{\mathbf{x}_0}|q_0) + D_{\varphi_1}(q_{\mathbf{x}_1}|q_1) \quad (6)$$

where $C$ is the matching cost and $D_{\varphi_0}$ and $D_{\varphi_1}$ are $\varphi$-divergences.

Using Propositions 1 and 2, we can show that the optimization from Eq. 3 is upper bounded by:

$$\alpha \, W_2^2(\hat{q}_{\mathbf{x}_1}, q_{\mathbf{x}_1}) - \beta \, \log \text{ESS}^*(q) \leq \beta \, D_2(q_0|q_{\mathbf{x}_0}) + \\ + \alpha \, L^2 \, \mathbb{E}_{(\mathbf{x}_0, \mathbf{x}_1) \sim q} \|\mathbf{x}_0 - \mathbf{x}_1\|^2 + \quad (7) \\ + \beta \, D_2(q_{\mathbf{x}_1}|q_1)$$

therefore, choosing $q$ via static unbalanced OT directly provides an upper bound for the efficiency vs approximation tradeoff cost.

In practice in many domains like docking, one cannot obtain many samples from each distribution, ruling out complete optimal transport coupling calculations, and therefore we also consider a simpler family of couplings that can be obtained with rejection sampling from individual independent samples from $q_0$ and $q_1$ but still maintains a bound on the transport cost: $q(\mathbf{x}_0, \mathbf{x}_1) \propto q_0(\mathbf{x}_0)q_1(\mathbf{x}_1)\mathbb{I}_{C(\mathbf{x}_0, \mathbf{x}_1) \leq \epsilon}$ for some predefined non-negative $\epsilon$.

# 4. Flexible Docking

In the flexible docking task, our goal is to learn the joint distribution over the bound structures (equivalently, poses thereof) of a protein-ligand complex given the distribution over the unbound structures. In many drug discovery applications, protein pocket locations are often known. We thus focus on the pocket-based flexible docking task, but emphasize that all components of our framework equivalently translate to the blind docking setting as well (when the protein pocket is unknown).

Ligand and protein poses can be regarded as elements of $\mathbb{R}^{3n_l}$ and $\mathbb{R}^{3n_p}$, where $n_l$ and $n_p$ are the number of atoms in the ligand and protein. However, during docking, ligand flexibility is largely concentrated in the torsion angles at rotatable bonds (Corso et al., 2022), while for proteins, the flexibility lies in the backbone frames (Jumper et al., 2021) and sidechain torsion angles. Motivated by the success of Intrinsic Diffusion Models (Corso, 2023) in similar domains, we reduce the space of ligand and protein poses by defining our generative model over these degrees of freedom.

While our couplings $q$ are defined based on costs $c$ in the Euclidean space, we posit that also implies equivalent couplings $q_{\mathbb{P}}$ between distributions on the product space $\mathbb{P}$ that largely governs docking flexibility.

## 4.1. Docking over Manifold Degrees of Freedom

For the distribution over ligand poses, we largely follow DIFFDOCK (Corso et al., 2022), learning a diffusion model over the product space of rotations, translations, and torsions, $\mathbb{P} = SO(3) \times \mathbb{R}^3 \times \mathbb{T}^{m_l}$. The key difference from DIFFDOCK is that our model accepts as input (for a diffusion time $t$), an intermediate protein structure governed by the choice of flow (see below) rather than a rigid structure.

To model conformational changes in protein structures upon docking, we employ our Unbalanced Flow Matching framework. The prior $q_0$ is defined as the distribution of computationally generated unbound structures (Lin et al., 2022), while the target distribution $q_1$ is defined over the crystallized bound structures. For both distributions, we only have access to samples thereof. For a protein with $n$ residues and $m_p$ sidechain torsion angles, we define the flow over the product space $SE(3)^n \times SO(2)^{m_p}$, where the $SE(3)$ frame for each residue corresponds to a roto-translation around the C$\alpha$ atom, and the hypertorus $\mathbb{T}^{m_p}$ over sidechain torsions. Designing a unbalanced FM objective then amounts to choosing a coupling $q(\mathbf{x}_0, \mathbf{x}_1)$, a conditional probability path $p_t(\mathbf{x}|\mathbf{x}_0, \mathbf{x}_1), (\mathbf{x}_0, \mathbf{x}_1) \sim q$, and the associated conditional vector field $\mathbf{u}_t(\mathbf{x}|\mathbf{x}_0, \mathbf{x}_1)$.

**Choice of coupling $q$.** A key requirement for $q$ is to be able to sample pairs during training. Because we typically only have access to one sample for the distribution over

bound structures (the crystal structure in PDB, typically unique), we cannot define $q$ via Unbalanced OT. Therefore, we approximate the optimal coupling with the distribution $q(\mathbf{x}_0, \mathbf{x}_1) \propto q_0(\mathbf{x}_0)q_1(\mathbf{x}_1)\mathbb{I}_{c(\mathbf{x}_0,\mathbf{x}_1)<c_{\text{dock}}}$, where $c(\mathbf{x}_0, \mathbf{x}_1)$ is defined as the aligned RMSD between the C$\alpha$ positions of the residues in the pocket and the neighborhood, and $c_{\text{dock}}$ is an empirically chosen cutoff to balance sample efficiency and mapping complexity. We can sample from $q$ by taking individual independent samples from $q_0$ and $q_1$ and rejecting if $c(\mathbf{x}_0, \mathbf{x}_1)) \geq c_{\text{dock}}$ ($c_{\text{dock}} = 4$ in our experiments).

**Flow Matching on $SE(3)$ and $\mathbb{T}$.** Following the disintegration of measures (Pollard, 2002), every $SE(3)$-invariant measure can be broken down into a $SO(3)$-invariant measure and a measure proportional to the Lebesgue measure on $\mathbb{R}^3$, allowing us to build flows independently on $SO(3)$ and $\mathbb{R}^3$. Following (Chen & Lipman, 2024), given two points $(\mathbf{x}_0, \mathbf{x}_1) \sim q$, the conditional probability path between $\mathbf{x}_0$ and $\mathbf{x}_1$ is given by the geodesic between them, $\mathbf{x} = \exp_{\mathbf{x}_0}(t \log_{\mathbf{x}_0}(\mathbf{x}_1))$, and the corresponding conditional flow is $\mathbf{u}_t(\mathbf{x}_t|\mathbf{x}_0, \mathbf{x}_1) = \frac{\log_{\mathbf{x}_t} \mathbf{x}_1}{1-t}$.

For $SO(3)$, the geodesics can be computed efficiently by using the axis-angle representation (equivalent to $\log(\mathbf{x}_1)$) and the parallel transport operation (left multiplication with $\mathbf{x}_0$), while $\exp$ is simply the matrix exponential. We view the torus $\mathbb{T}$ as the quotient space $\mathbb{R}/2\pi\mathbb{Z}$, thus $\exp_{\mathbf{x}_0}(\mathbf{x}_1) = (\mathbf{x}_0 + \mathbf{x}_1) \mod 2\pi$ (equivalent to wrapping around $\mathbb{R}$), and $\log_{\mathbf{x}_0} \mathbf{x}_1 = \arctan 2(\sin(\mathbf{x}_1 - \mathbf{x}_0), \cos(\mathbf{x}_1 - \mathbf{x}_0))$.

## 4.2. Training and Inference

**Manifold Docking.** Although the flow and diffusion objectives for protein and ligand poses are defined on the respective product spaces, our training and inference procedures are designed to operate on 3D coordinates directly, allowing the model to learn better, and generalize to unseen complexes. (Jing et al., 2022; Corso et al., 2022). For the torsion angles in the sidechains and the ligand, we apply a conformer matching procedure (Jing et al., 2022; Plainer et al., 2023), to avoid a distribution shift (in terms of local structures), between training and inference.

**Confidence Model.** The confidence model can be thought of as reweighting samples from the learned flow $\hat{q}_{\mathbf{x}_1}$ in accordance with the true marginal $q_1$. To collect training data for the confidence model, we use a smaller version of mthe anifold docking model to generate 20 poses per complex, which are then assigned a label based on whether the predicted ligand and protein pocket poses have RMSDs below 2Å and 1Å respectively. The confidence model is then trained with cross entropy loss. During inference, we generate poses in parallel with our manifold docking model, which are then scored by the confidence model.

*Table 2.* **Top-1 PDBBind ESMFold Docking Performance**. Percentage of predictions with ligand RMSD < 2Å and All-Atom RMSD < 1Å and median RMSDs. In parenthesis, we specify number of sampled poses. For RE-DOCK, values marked with ∗ indicate that we could not compute those values, and used the closest reported numbers.

| Method | Ligand RMSD | | All-Atom RMSD | | Runtime (s) |
|---|---|---|---|---|---|
| | Median ↓ | % < 2Å ↑ | Median ↓ | % < 1Å ↑ | |
| SMINA (rigid) | 7.7 | 6.6 | - | - | 258 |
| SMINA | 7.3 | 3.6 | 1.7 | 5.2 | 1914 |
| GNINA (rigid) | 7.5 | 9.7 | - | - | 260 |
| GNINA | 7.2 | 6.6 | 1.7 | 4.5 | 1575 |
| DIFFDOCK (pocket, rigid) (40) | 2.6 | 37.8 | - | - | 61 |
| DIFFDOCK-POCKET (10) | 2.6 | 41.0 | 1.4 | 31.6 | 17 |
| DIFFDOCK-POCKET (40) | 2.6 | 41.7 | 1.3 | 32.1 | 61 |
| REDOCK (10) | 2.5 | 39.0 | 1.2 | 39.8∗ | 15 |
| REDOCK (40) | 2.4 | 42.9 | 1.2 | 38.4∗ | 58 |
| FLEXDOCK (10) | 2.6 | 39.5 | 1.2 | 44.1 | 10 |
| FLEXDOCK (40) | 2.5 | 40.8 | 1.1 | 43.9 | 38 |

## 5. Experiments

**Data.** We train our models on the PDBBind dataset (Liu et al., 2017), using the time-based split, and validate on the PDBBind and PoseBusters (Buttenschoen et al., 2024) benchmark datasets. We computationally generated structures from ESMFOLD (Lin et al., 2022) as samples from the distribution of unbound structures.

**Baselines.** For PDBBind, we compare FLEXDOCK, with state-of-the-art search-based methods SMINA and GNINA, ML-based pocket level docking methods in DIFFDOCK-POCKET (Plainer et al., 2023) and RE-DOCK (Huang et al., 2024). On the PoseBusters benchmark dataset, we also compare against recent publicly available *co-folding* methods – ROSETTAFOLD-ALLATOM (Krishna et al., 2024) and UMOL (Bryant et al., 2023).

*Table 3.* **Top-1 PoseBusters Docking Performance**. Percentage of predictions with ligand RMSD < 2Å. ∗ assume knowledge of holo structure. † blind docking. # trained on significantly more data from the whole PDB.

| Method | Ligand RMSD |
|---|---|
| | % < 2Å ↑ |
| GOLD∗ | 58 |
| VINA∗ | 60 |
| DEEPDOCK∗† | 20 |
| DIFFDOCK∗† | 38 |
| ROSETTAFOLD-ALLATOM†# | 42 |
| UMOL | 45 |
| FLEXDOCK (10) | 46 |

**Metrics.** We evaluate the quality of both the predicted ligand and pocket atom poses. The quality of predicted structures is measured by the heavy-atom RMSDs to the ground truth structures. For PoseBusters, we only report the docking accuracy as measured by % of ligand RMSDs < 2Å. Additional details regarding the experimental setup, data and baselines can be found in Appendix E.

**Results.** On the PDBBind dataset, FLEXDOCK achieves strong improvements on predicting protein conformations (All-Atom RMSD < 1Å), while retaining comparable docking accuracy (ligand RMSD < 2Å) and faster runtimes. On the PoseBusters dataset, FLEXDOCK achieves better performance than many co-folding methods, despite being trained only on the PDBBind dataset.

## 6. Conclusion

We propose Unbalanced Flow Matching, a generalization of Flow Matching that allows us to relax the marginal constraints and learn simpler flows. We theoretically analyze the tradeoffs between sample efficiency and approximation capabilities these relaxations induce. Empirically, we validate our framework on flexible docking, with strong improvements in modelling protein conformational changes, while retaining comparable docking accuracy.

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

# A. Propositions

Note: in all derivations and definitions in this section, we will assume that the distributions we work with are defined in Euclidean space and have full support.

## A.1. Unbalanced FM Approximation Error

**Lemma 1.** *Given $\mathbf{X}$ a random vector and $\mathbf{c}$ a constant vector, we have:*

$$\mathbb{E}\left[\|\mathbf{X} - \mathbf{c}\|^2\right] \geq \|\mathbb{E}[\mathbf{X}] - \mathbf{c}\|^2$$

*Proof.* Expanding the left-hand side:

$$\mathbb{E}\left[\|\mathbf{X} - \mathbf{c}\|^2\right] = \mathbb{E}\left[\sum_{i=1}^{n}(X_i - c_i)^2\right] = \sum_{i=1}^{n}\mathbb{E}\left[(X_i - c_i)^2\right]$$

For any random variable $X_i$,

$$\mathbb{E}\left[(X_i - c_i)^2\right] = \mathbb{E}\left[(X_i - \mathbb{E}[X_i] + \mathbb{E}[X_i] - c_i)^2\right]$$

$$=\mathbb{E}\left[(X_i-\mathbb{E}[X_i])^2+2(X_i-\mathbb{E}[X_i])(\mathbb{E}[X_i]-c_i)+(\mathbb{E}[X_i]-c_i)^2\right]$$

Since $\mathbb{E}[X_i - \mathbb{E}[X_i]] = 0$, the middle term vanishes:

$$= \mathbb{E}[(X_i - \mathbb{E}[X_i])^2] + (\mathbb{E}[X_i] - c_i)^2$$

Therefore:

$$\mathbb{E}\left[(X_i - c_i)^2\right] = \text{Var}(X_i) + (\mathbb{E}[X_i] - c_i)^2$$

Summing over all components:

$$\mathbb{E}\left[\|\mathbf{X} - \mathbf{c}\|^2\right] = \sum_{i=1}^{n}\text{Var}(X_i) + \sum_{i=1}^{n}(\mathbb{E}[X_i] - c_i)^2$$

$$= \sum_{i=1}^{n}\text{Var}(X_i) + \|\mathbb{E}[\mathbf{X}] - \mathbf{c}\|^2$$

Since $\text{Var}(X_i) \geq 0$ for each component $i$, we have our result.

$\square$

**Assumption 2** (Existence and uniqueness of smooth flows) *For each $\mathbf{x} \in \mathbb{R}^d$ and $s \in [0,1]$ there exist unique flows $(Y_{s,t}^{\mathbf{x}})_{t\in[s,1]}$ and $(Z_{s,t}^{\mathbf{x}})_{t\in[s,1]}$ starting in $Y_{s,s}^{\mathbf{x}} = \mathbf{x}$ and $Z_{s,s}^{\mathbf{x}} = \mathbf{x}$ with velocity fields $v_\theta(\mathbf{x},t)$ and $v^X(\mathbf{x},t)$ respectively. Moreover, $Y_{s,t}^{\mathbf{x}}$ and $Z_{s,t}^{\mathbf{x}}$ are continuously differentiable in $\mathbf{x}$, $s$ and $t$.*

**Assumption 3** (Regularity of approximate velocity field) *The approximate flow $v_\theta(\mathbf{x}, t)$ is differentiable in both inputs. Also, for each $t \in (0,1)$ there is a constant $L_t$ such that $v_\theta(\mathbf{x},t)$ is $L_t$-Lipschitz in $\mathbf{x}$.*

**Proposition 1.** *Under assumptions 2 and 3 from Benton et al. (2023) reported above, we have:*

$$W_2^2(\hat{q}_{\mathbf{x}_1}, q_{\mathbf{x}_1}) \leq L^2 \cdot \mathbb{E}_{(\mathbf{x}_0,\mathbf{x}_1)\sim q}\|\mathbf{x}_0 - \mathbf{x}_1\|^2 \quad (8)$$

$$where \ L = \exp\left[\int_0^1 L_t dt\right]. \quad (9)$$

*Proof.* Let $\mathbf{u}_t(\cdot)$ and $\mathbf{v}_t(\cdot; \theta)$ be the marginal vector fields generating $\psi_t$ and $\hat{\psi}_t$ respectively.

Using Theorem 1 from Benton et al. (2023) we have:

$$W_2^2(\hat{q}_{\mathbf{x}_1}, q_{\mathbf{x}_1}) \leq L^2 \int_0^1 \mathbb{E}_q[\|\mathbf{u}_t(\mathbf{X}_t) - \mathbf{v}_t(\mathbf{X}_t; \theta)\|^2]dt$$

For any $\mathbf{x}_t$, we can use Lemma 1 and the knowledge that $\mathbf{u}_t(\mathbf{x}_t) = \mathbb{E}_{q|\mathbf{X}_t=\mathbf{x}_t}\mathbf{u}_t(\mathbf{x}_t|\mathbf{x}_0, \mathbf{x}_1)$:

$$\|\mathbf{u}_t(\mathbf{x}_t)-\mathbf{v}_t(\mathbf{x}_t;\theta)\|^2 \leq \mathbb{E}_{q|\mathbf{X}_t=\mathbf{x_t}}\left[\|\mathbf{u}_t(\mathbf{x}_t|\mathbf{x}_0,\mathbf{x}_1) - \mathbf{v}_t(\mathbf{x}_t;\theta)\|^2\right]$$

therefore:

$$W_2^2(\hat{q}_{\mathbf{x}_1}, q_{\mathbf{x}_1}) \leq L^2 \int_0^1 \mathbb{E}_q[\|\mathbf{u}_t(\mathbf{x}_t|\mathbf{X}_0,\mathbf{X}_1)-\mathbf{v}_t(\mathbf{X}_t;\theta)\|^2]dt$$

the expression inside the square root is our loss function that we are trying to minimize, therefore, under the assumption that the zero function is in our functional space, we can say:

$$W_2^2(\hat{q}_{\mathbf{x}_1}, q_{\mathbf{x}_1}) \leq L^2 \int_0^1 \mathbb{E}_q[\|\mathbf{u}_t(\mathbf{x}_t|\mathbf{X}_0,\mathbf{X}_1)\|^2]dt$$

$$= L^2 \int_0^1 \mathbb{E}_q[\|\mathbf{X}_1 - \mathbf{X}_0\|^2]dt$$

$$= L^2 \cdot \mathbb{E}_{(\mathbf{x}_0,\mathbf{x}_1)\sim q}\|\mathbf{x}_0 - \mathbf{x}_1\|^2$$

$\square$

## A.2. Unbalanced FM Sample Efficiency

**Proposition 2.** *The effective sample size, $ESS^*$, for sampling $q_1$ when having access to samples of $q_0$ and a perfectly trained flow with coupling distribution $q$ is bounded by:*

$$ESS^*(q) \geq \exp\left[-D_2(q_0|q_{\mathbf{x}_0}) - D_2(q_{\mathbf{x}_1}|q_1)\right] \quad (10)$$

*where $D_2$ is the Rényi Divergence of order 2.*

*Proof.* When comparing pairs of distributions $p$ and $q$ the (population) effective sample size $\mathrm{ESS}^*(p, q)$ is defined as (Maia Polo & Vicente, 2023):

$$\mathrm{ESS}^*(p, q) := \exp[-D_2(p|q)]$$

and it can be considered as the percentage of effective samples from $q$ one can obtain when taking samples from $p$.

Similarly, we define $\mathrm{ESS}^*(q)$ in our setting as the percentage of effective samples from $q_1$ one can obtain from $q_0$ using $\psi_1$. While $\psi_1$ could be directly applied to any distribution, including $q_0$, it is hard to model its pushforward analytically. On the other hand, we know that $q_{\mathbf{x}_1}$ is the pushforward of $q_{\mathbf{x}_0}$, therefore we can obtain samples from $q_1$ by (1) reweighting samples of $q_0$ into samples of $q_{\mathbf{x}_0}$, (2) transporting samples from $q_{\mathbf{x}_0}$ to samples of $q_{\mathbf{x}_1}$ and (3) reweighting samples from $q_{\mathbf{x}_1}$ into samples from $q_1$. By assumption of perfect flow step (2) has perfect efficiency, however, steps (1) and (3) both may require more than one sample in expectation to be unbiased. This translates into an efficiency equal to the product of the two effective sample sizes:

$$\mathrm{ESS}^*(q) \geq \mathrm{ESS}^*(q_0, q_{\mathbf{x}_0}) \, \mathrm{ESS}^*(q_{\mathbf{x}_1}, q_1) =$$
$$\exp\left[-D_2(q_0|q_{\mathbf{x}_0})\right] \, \exp\left[-D_2(q_{\mathbf{x}_1}|q_1)\right]$$
$$= \exp\left[-D_2(q_0|q_{\mathbf{x}_0}) - D_2(q_{\mathbf{x}_1}|q_1)\right].$$

where the inequality derives from the possibility of the existence of more effective procedures for this sampling that do not require passing from samples of $q_{\mathbf{x}_0}$ and $q_{\mathbf{x}_1}$.

$\square$

## B. Background and Related Work

**Flow matching (FM)** (Lipman et al., 2022; Albergo et al., 2023) is a generative modeling paradigm that was introduced as a flexible generalization of diffusion models, and allows learning a transport between arbitrary distributions with a simulation-free objective. Given two distributions $q_0$ and $q_1$, FM provides a way of learning a vector field $v_t$ which induces a continuous normalizing flow $\psi_t(x)$ that transports $q_0$ to $q_1$, i.e., $q_1(x) = [\psi_1]_\# q_0(x))$, where $\#$ denotes the pushforward operator.

The key idea in FM is defining a conditional flow $\psi_t(x_0|x_1)$ interpolating between $x_0 \sim q_0$ and $x_1 \sim q_1$, and its associated vector field $u_t(x_t|x_1) = \frac{d}{dt}\psi_t(x_t|x_1)$. One can then learn the marginal vector field $\hat{v}_t(x, t; \theta)$ with a neural network ($\hat{v}_t(x, t; \theta) \approx v_t(x, t)$, by regressing against the conditional vector field with the CFM objective:

$$\mathcal{L}_{\mathrm{CFM}} = \mathbb{E}_{t, x_0 \sim q_0, x_1 \sim q_1} \|v_t(x_t; \theta) - u_t(x_t|x_1)\|^2 \quad (11)$$

FM was further generalized by Pooladian et al. (2023) and Tong et al. (2023) which showed that the sampling distribution in the CFM objective, which we will refer to as coupling

distribution, does not have to be independent samples from $q_0$ and $q_1$ and can be an arbitrary joint distribution $q(x_0, x_1)$ as long as it satisfies the marginal constraints being $q_0$ and $q_1$ respectively. This formulation enabled drawing a connection between FM and optimal transport (OT). When using OT to define the coupling distribution $q$, the flows become straight and the transport cost $\mathbb{E}_{q_0(x_0)} \|\psi_1(x_0) - x_0\|^2$ is the OT cost $W_2^2(q_0, q_1)$.

**Protein-ligand binding** When proteins bind to small molecules, their structural distribution adjusts to fit the molecule. Understanding this conformational change is critical for accurately predicting binding interactions, and computational methods for this fall into two categories: *co-folding* and *flexible docking*. Co-folding involves predicting the bound structure of the protein and the ligand from scratch as a single task. Based on the success of AlphaFold 2 (AF2) (Jumper et al., 2021) for protein structure prediction, a number of methods have extended AF2 for small-molecule co-folding (Qiao et al., 2024; Bryant et al., 2023; Krishna et al., 2024; Abramson et al., 2024). While these have achieved varied success, they typically require large amounts of training data and have slow inference times. (Wang et al., 2023) adopts a *co-folding* strategy based on diffusion processes, but instead of the Euclidean space, the diffusion processes are defined on the backbone torsion angles of the protein, and the product space of rotations, translations and torsions for the ligand.

**Protein Conformational Changes and Flexible docking** Flexible docking assumes access to unbound structures of proteins (known as apo) and predicts how these will change upon ligand binding (producing holo-structures). Since the conformational change is usually small and localized due to the molecule's size and energetic impact, this approach has been preferred for protein-ligand structure prediction, making it suitable for large-scale screening pipelines.

Traditional *search-based* docking methods define a scoring function and search the space of possible poses (through rigid movement and torsion angle changes of the ligand) to find the minimum of the scoring function (Alhossary et al., 2015; McNutt et al., 2021). These methods can typically, incorporate protein flexibility by adding torsion angles of the sidechains in the pocket to the search space. However, due to the increased dimensionality and the protein's flexibility beyond the sidechains, traditional methods struggle to find optimal joint poses.

Recently, a number of deep learning methods have been proposed that leverage the flexibility of proteins in molecular docking. DIFFDOCK-POCKET (Plainer et al., 2023) and RE-DOCK (Huang et al., 2024) use diffusion and diffusion bridge models to model the flexibility in protein pocket sidechains in addition to ligand flexibility. DYNAMICBIND

(Lu et al., 2024), the closest related work to ours, incorporates backbone flexibility, with hardcoded noise perturbation rules that interpolate from apo residue frames to holo residue frames. While the operate in the blind docking setting, they do not directly model the atomic positions, leaving limited utility for downstream applications such as binding affinity calculation. (Somnath et al., 2023) explicitly predicts the conformational changes between apo and holo states of proteins using diffusion Schrödinger Bridges, by treating the apo and holo states as paired data. While they account for the constraint that minibatch-OT maps cannot be computed as in Pooladian et al. (2023), directly interpolating between conformational states suffer from the same complex learning task as we outlined in Section 3.2. Furthermore, their method was not evaluated on the flexible docking task.

## C. Training and Inference

In this section, we present the training and inference procedures for our manifold docking (Algoritm C, C). We refer to unbound protein structures as apo structures, and the bound structures as holo structures. Recall that our goal is to learn a distribution over holo structures, given the apo structure and a seed conformation of the ligand. Similar to (Corso et al., 2022), we are in a setting, where traditional generative modeling where one has access to multiple samples from the same data distribution, we only have a single $(\mathbf{x}^*, \mathbf{y}_{\text{apo}}, \mathbf{y}_{\text{holo}})$ per protein-ligand complex. This implies that the training loop (Algorithm C) now proceeds over different distributions, along with a single sample from that distribution. This sample is then accepted or rejected depending on the cutoff $c_{\text{dock}}$, thus inducing an unbalanced coupling and flow.

**Pocket Extraction**   As our focus is on the flexible protein docking task, we first extract the protein pocket given apo and holo structures. We define the pocket residues as all residues in the holo structure that have atleast one heavy atom within 5Å of any ligand atom. Given these pocket residues, the pocket center is defined based on the positions of the C$\alpha$ atom in the apo structure. To construct the geometric graphs (Appendix E), we also use the residues which have a C$\alpha$ atom within 20Å of the pocket center. This additional buffer is added to improve the model's robustness to exact pocket definitions, and also add geometric information from the pocket neighborhood.

**Aligning Apo-Holo Frames**   A residue frame (Jumper et al., 2021), is characterized by a tuple $(R, t) \in SE(3)$, where the rotation $R$ is about the origin of the residue, and $t$ specifies the position of the C$\alpha$ atom. Before applying the conformer matching step (explained below) to the protein sidechains, we align the frames of the apo and holo structures, by computing the rotation that aligns the $N - C\alpha$

vectors of the corresponding residues. The alignment will not be perfect owing to differences in the bond lengths and bond angles between the computationally generated and ground truth structures, but provides the closest modification of the apo structure backbone to the holo structure one.

**Conformer Matching.**   For both the ligand and the protein sidechains, we apply the conformer matching procedures in (Jing et al., 2022) and (Plainer et al., 2023), where, given the local structures from computational methods, we find the closest (in a RMSD sense) structure to the ground truth by modifying the appropriate torsion angles. The conformer matching procedure is employed to prevent a distribution shift between training and inference in the local structures that are considered rigid in the manifold docking process. To elaborate, the local structures (such as bond lengths and bond angles) vary between RDKit (for ligands) and ESM-Fold (for proteins) generated structures, and their ground truth counterparts. If we train our models with ground truth local structures, this would cause a distribution shift at inference time, when we only have access to local structures, provided by RDKit and ESMFold.

## D. Model Architecture

We use message passing networks based on tensor products of irreducible representations (irreps) of $SO(3)$, implemented with the e3nn library.

**Graph Construction.**   We represent structures as geometric heterogenous graphs, with nodes comprising ligand heavy atoms, receptor residues in the pocket and neighborhood (located at the position of C$\alpha$ atoms), and the heavy atoms of the pocket residues. We chose to only model the heavy atoms of the pocket residues for two reasons - i) this provides a useful sparsity constraint for computational and memory efficiency, and ii) typically, most of the conformational changes in the protein involve the pocket atoms, and modelling this explicitly would facilitate downstream applications such as affinity prediction. We also adopt different cutoffs depending on the types of nodes being connected, largely following (Corso et al., 2022):

1. Ligand atoms-ligand atoms, receptor atoms-receptor atoms, and ligand atom-receptor atom interactions use a cutoff of 5Å. Covalent bonds between ligand atoms are explicitly modelled with initial edge embeddings to reflect the type of bond. For receptor atoms, we limit the maximum number of neighbors to 12.

2. For receptor residue interactions, we use a distance cutoff of 15Å, with a maximum neighbor limit of 24.

3. For interactions between ligand atoms and receptor residues, unlike (Corso et al., 2022), we found using

---

**Algorithm 1** TRAINING EPOCH: MANIFOLD DOCKING

---

**Input:** Training Pairs $\{(\mathbf{x}^*, \mathbf{y}_{\text{apo}}, \mathbf{y}_{\text{holo}})\}$; RDKit predictions $\{\mathbf{c}\}$, RMSD cutoff $c_{\text{dock}}$
**Input:** Pocket radius $r$; Pocket Buffer $b$
**Input:** C$\alpha$ operator $[\cdot]_{C\alpha}$

**foreach** $\mathbf{c}, \mathbf{x}^*, \mathbf{y}_{apo}, \mathbf{y}_{holo}$ **do**
    Let $\mathbf{x}_0 \leftarrow \arg\min_{\mathbf{x}} \text{RMSD}(\mathbf{x}^*, \mathbf{x})$
    $\mathbf{y}_{\text{center}}, \{i\}_{\text{pocket}} = \text{EXTRACTPOCKET}(\mathbf{y}_{\text{apo}}, \mathbf{y}_{\text{holo}}, r, b)$
    $\mathbf{y}_{\text{apo}} \leftarrow \text{RMSDALIGN}(\mathbf{y}_{\text{apo}}, \mathbf{y}_{\text{holo}}, \{i\}_{\text{pocket}})$
    **if** $\text{RMSD}(\mathbf{y}_{apo}, \mathbf{y}_{holo}) > c_{dock}$ **then**
       | **continue**
    **else**
       $\mathbf{y}_{\text{apo}}^{\text{FA}}, \Delta R^{\text{bb}} \leftarrow \text{FRAMEALIGN}(\mathbf{y}_{\text{apo}}, \mathbf{y}_{\text{holo}})$
       $\mathbf{y}_{\text{apo}}^{\text{FA,SC}}, \Delta\theta^{\text{sc}} \leftarrow \text{SCCONFMATCH}(\mathbf{y}_{\text{apo}}^{\text{FA}}, \mathbf{y}_{\text{holo}})$
       Sample $t \sim \mathcal{U}(0,1)$

       `// Ligand Diffusion`
       Sample $\Delta r, \Delta R, \Delta\theta$ from diffusion kernels $p_t^{\text{tr}}(\cdot|0), p_t^{\text{rot}}(\cdot|0), p_t^{\text{tor}}(\cdot|0)$
       Compute $\mathbf{x}_t$ by applying $(\Delta r, \Delta R, \Delta\theta)$ to $\mathbf{x}_0$

       `// Protein Flow`
       $t^{\text{sc}}, t_{\text{rot}}^{\text{bb}}, t_{\text{tr}}^{\text{bb}} = \text{COMPUTETIME}(t, \alpha^{\text{sc}}, \alpha_{\text{rot}}^{\text{bb}}, \alpha_{\text{tr}}^{\text{bb}})$
       Interpolate $\Delta r_t^{\text{bb}} \leftarrow [\mathbf{y}^{apo}]_{C\alpha} \cdot (1-t) + [\mathbf{y}^{holo}]_{C\alpha} \cdot t$
       $u_t^{\text{tr,bb}}(\cdot|z) \leftarrow [\mathbf{y}^{holo}]_{C\alpha} - [\mathbf{y}^{apo}]_{C\alpha}$

       Interpolate $\Delta R_t^{\text{bb}} \leftarrow \exp\left(t_{\text{rot}}^{\text{bb}} \log(\Delta R^{\text{bb}})\right)$
       $u_t^{\text{rot,bb}}(\cdot|z) \leftarrow \frac{\log_{\Delta R_t^{\text{bb}}}(\Delta R^{\text{bb}})}{1 - t_{\text{rot}}^{\text{bb}}}$

       Interpolate $\Delta\theta_t^{\text{sc}} \leftarrow \exp\left(t^{\text{sc}} \log(\Delta\theta^{\text{sc}})\right)$
       $u_t^{\text{sc}}(\cdot|z) \leftarrow \frac{\log_{\Delta\theta_t^{\text{sc}}}(\Delta\theta^{\text{sc}})}{1 - t^{\text{sc}}}$

       Compute $\mathbf{y}_t$ by applying $\left(\Delta r^{\text{bb}}, \Delta R^{\text{bb}}, \Delta\theta^{\text{sc}}\right)$ to $\mathbf{y}_{\text{apo}}$

       Predict scores and drifts $\alpha, \beta, \gamma, \delta, \epsilon, \eta \leftarrow s(\mathbf{x}_t, \mathbf{y}_t, t)$

       `// Ligand Loss`
       $\mathcal{L}_{\text{lig}} = \|\alpha - \nabla\log p_t^{\text{tr}}(\cdot|0)\|^2 + \|\beta - \nabla\log p_t^{\text{rot}}(\cdot|0)\|^2 + \|\gamma - \nabla\log p_t^{\text{tor}}(\cdot|0)\|^2$
       `// Protein Loss`
       $\mathcal{L}_{\text{prot}} = \|\delta - u_t^{\text{tr,bb}}(\cdot|z)\|^2 + \|\epsilon - u_t^{\text{rot,bb}}(\cdot|z)\|^2 + \|\eta - u_t^{\text{sc}}(\cdot|z)\|^2$
       Apply optimization step on $\mathcal{L} = \mathcal{L}_{\text{prot}} + \mathcal{L}_{\text{lig}}$
    **end**
**end**

---

---

**Algorithm 2** INFERENCE: MANIFOLD DOCKING

---

**Input:** RDKit predictions $\{\mathbf{c}\}$, Apo structure $\mathbf{y}_{\text{apo}}$ of the protein pocket
**Input:** Inference Steps $N$
Sample $\Theta_N \sim \mathcal{U}(SO(2)^m)$, $R_N \sim \mathcal{U}(SO(3))$, $r_n \sim \mathcal{N}(0, \sigma_{\text{tr}}^2)$
  Apply $\Theta_N, R_N, r_n$ to $c$ to get $\mathbf{x}_N$
  Set $\mathbf{y}_N \leftarrow \mathbf{y}_{\text{apo}}$
  $\Delta t \leftarrow 1/N$
  **for** $n \leftarrow N$ **to** $1$ **do**
  $\quad$ $t \leftarrow n/N$
  $\quad$ Predict scores and drifts $\alpha, \beta, \gamma, \delta, \epsilon, \eta \leftarrow s(\mathbf{x}_n, \mathbf{y}_n, t)$

  $\quad$ // Ligand Updates
  $\quad$ $\Delta\sigma_{\text{tr}}^2 = \sigma_{\text{tr}}^2(n/N) - \sigma_{\text{tr}}^2((n-1)/N)$
  $\quad$ $\Delta\sigma_{\text{rot}}^2 = \sigma_{\text{rot}}^2(n/N) - \sigma_{\text{rot}}^2((n-1)/N)$
  $\quad$ $\Delta\sigma_{\text{tor}}^2 = \sigma_{\text{tor}}^2(n/N) - \sigma_{\text{tor}}^2((n-1)/N)$
  $\quad$ Sample $\mathbf{z}_{\text{tr}}, \mathbf{z}_{\text{rot}}, \mathbf{z}_{\text{tor}}$ from $\mathcal{N}(0, \sigma_{\text{tr}}^2), \mathcal{N}(0, \sigma_{\text{rot}}^2), \mathcal{N}(0, \sigma_{\text{tor}}^2)$
  $\quad$ Apply $(\alpha, \beta, \gamma)$ to $\mathbf{x}_n$ to get $\mathbf{x}_{n-1}$

  $\quad$ // Protein Updates
  $\quad$ $\Delta r_n^{\text{bb}} \leftarrow \delta \cdot \Delta t$
  $\quad$ $\Delta R_n^{\text{bb}} \leftarrow \epsilon \cdot \Delta t$
  $\quad$ $\Delta\theta_n^{\text{sc}} \leftarrow \eta \cdot \Delta t$
  $\quad$ Apply $\left(\Delta r_n^{\text{bb}}, \Delta R_t^{\text{bb}}, \Delta\theta_n^{\text{sc}}\right)$ to $\mathbf{y}_n$ to get $\mathbf{y}_{n-1}$
  **end**

---

the dynamic cutoff based on the ligand translation noise to cause NaNs during training, possibly due to missing connections. We thus used distance cutoff of 80Å between ligand atoms and receptor residues.

4. Receptor pocket atoms are also connected to their corresponding residues.

**Featurization** We adopted the same featurization as DIFF-DOCK, using the residue type and the embeddings with ESM2 Language model for the residues, the atom type and other chemical properties for the ligand and receptor atoms.

**Manifold Docking** We retain the core architecture of DIFFDOCK (Corso et al., 2022), with the tensor product convolution based message-passing layers, followed by a convolution with the center of mass to predict the rotational and translation scores for the ligand. For the torsion angles in the ligand and sidechain torsion angles in the protein, we use the pseudotorque layer from (Jing et al., 2022), adapted accordingly for the sidechains. To predict the rotation and translation flows for the residues (which are $SE(3)$ equivariant), we use a linear layer that transforms the irreps of the residue embeddings to a single odd and even vector (one for each flow). As the residues constitute a coarse-grained representation of the protein, we sum the odd and even vector representations to obtain the predictions. The magnitudes of the predictions are then adjusted with an MLP.

**Confidence Model** The embedding layers for the confidence model follow the same architecture as for manifold docking. The aggregated ligand, receptor residue, and receptor atom embeddings are concatenated, and updated with an MLP to predict the final confidence (a $SE(3)$ invariant output).

## E. Experimental Details

**Data** For training our models, we use the PDBBind dataset (Liu et al., 2017) whose complexes were extracted from the PDB. Following (Stärk et al., 2022; Corso et al., 2022), we adopt the time-based split of PDBBind, where the 17k complexes before 2019 were divided into training and validation sets, while the 363 complexes after 2019 form the test set. We download the PDBBind data as it is provided by EquiBind from https://zenodo.org/record/6408497. These files are first processed by PDBFixer from the OpenMM toolbox (Eastman et al., 2017), to replace non standard residues and add missing atoms. We then used the PDBFixer processed files to extract the protein sequence, and predict its structure with ESMFold (Lin et al., 2022). The ESMFold generated files are also processed by PDBFixer to add missing atoms such as terminal oxygens, at the end of a chain. These processed files now constitute our apo structures, while the processed analogues from PDBBind constitute our holo structures. We further remove

hydrogen atoms while aligning the apo and holo structures.

For inference, we also use the PoseBusters benchmark dataset (Buttenschoen et al., 2024), a carefully-selected set of structures from the PDB. PoseBusters consists of crystal structures released since 2021 (no overlap with the PDBBind training set), which are subject to several quality control filters followed by a final sequence-based clustering, resulting in 428 complexes. We adopt the same strategies with `PDBFixer` for processing the PoseBusters files, followed by the generation of ESMFold structures.

**Metrics**    To evaluate the generated ligand and protein pocket poses, we compute the RMSD between the predicted and ground truth poses after alignment. This alignment is computed based on the Kabsch alignment between the atoms in the protein pocket, in the ground truth and predicted poses. To account for permutation symmetries in the ligand, we use the symmetry-corrected RMSD of sPyRMSD. For the ligand, besides the median RMSD, we report the % of RMSDs below 2Å, which is a commonly adopted metric for judging the quality of docking predictions (Alhossary et al., 2015; Hassan et al., 2017; McNutt et al., 2021). For the protein pocket atoms, besdies the median RMSD, we report the % of RMSDs below 1Å, where we chose the 1Å cutoff, typically treated as atomic accuracy.

**Training Details**    For our manifold docking model (75.3 M parameters), we use an exponential moving average of weights (EMA) during training, which is updated every optimization step, with a decay factor of $0.999$. We train the model on $4$ RTX A6000 GPUs, with a batch size of $4$ per GPU. Every 10 epochs, we run inference for 20 steps with the EMA weights on 500 complexes in the validation set, and save the model with the largest percentage of ligand RMSDs < 2Å. The initial learning rate of the model is $0.001$, which is updated with a learning rate scheduler with decay $0.7$ if the percentage of complexes with ligand RMSDs < 2Å does not improve over 30 epochs. We train our model for 600 epochs, after which we did not observe a noticeable increase in ligand RMSDs < 2Å metric. We use the ADAM optimizer for all our models.

For the confidence model, we use a smaller version of the manifold docking model 4 M parameters to generate 20 poses (both ligand and protein) per training complex. For the ligand, we assign label 1 if the RMSDS between predicted (after alignment) and ground truth pose is <2Å, while for protein pocket atoms, we adopt a RMSD cutoff of 1Å. We train the confidence model for around 100 epochs, and save the model with the best accuracy. We found the model predicting only the ligand pose confidence to offer the best tradeoff between ligand and pocket atom prediction confidence.

**Runtimes**    Similar to other ML docking baselines, we measure runtimes for the manifold docking and confidence model. These runtimes are calculated on a single NVIDIA A100-80GB GPU, with the preprocessing steps entailing ESM2 embedding generation and RDKit conformer generation. The geometric graphs are generated on the fly as part of the model and thus already included in the runtimes.