# OpenReview forum: "Flexible Docking via Unbalanced Flow Matching"
_ICML.cc/2024/Workshop/ML4LMS — ML4LMS Poster_

### Official Review · Reviewer_N5XR · 2024-06-12

**Rating:** 7
**Confidence:** 4

**Review:**

### **Summary**

The paper tackle the problem of flexible protein-ligand docking where the protein is not kept rigid. The work leverages flow matching to model the task as a generative problem; however, they show that the standard flow matching method faces a complicated learning task that results in poor performance. Typically, optimal couplings of samples between the prior and target distribution results in an easier learning task; however, in the case of protein-ligand docking, there are cases where there is only a single sample from the target distribution, rendering OT ineffective. To remedy this, they propose unbalanced flow matching that incorporates rejection sampling to approximate optimal couplings between samples from the prior and samples from the data distribution. This results in shorter path lengths for flow matching, achieving competitive performance on $\leq 2A$ docking task and better performance on $\leq 1A$ docking task compared to previous methods.

### **Strengths**

- The paper is well-written; motivation and method are clearly explained.
- Simple, yet effective approach with proofs to support correctness.

### **Weaknesses**

- Lack of results for the confidence model.

### **Questions**

1. Section 2.1, lines 104-108

    > Secondly, even if one bypasses the issue of single samples
    from the bound structures during training through expensive
    methods like NMR or extended molecular dynamics, the
    OT cost will likely remain very high.
    >

    Why would the OT cost remain high? My understanding is the protein conformation does not change much after binding; hence, the coupling via OT will provide “optimal couplings” but samples from $q_1$  will largely be the same.

2. In Appendix C, subsection Conformer Matching: Does the conformer matching act similarly to optimal coupling for the case of ligands (and also for the protein sidechains)?
3. How is the method’s performance concerning physically-plausible structures? What is the percentage of steric clashes? Reference to a study done in DiffDock [1] Appendix F.
4. How is the performance of the confidence model? It would be nice to see a similar experiment as done in DiffDock [1] Figure 3.
5. How does the method compare to DynamicBind [2]?

### **Comments**

- It would be interesting to see results on the new DockGen benchmark [3].
- minor spelling corrections
    - Section 3.2, subsection Confidence Model

        “of mthe anifold docking model” → “of the manifold docking model”

    - Appendix, subsection C Training and Inference, line 514

        (Algoritm C, C) → (Algorithm C)


### **Summary of Review**

The work presents a simple yet effective method construct a coupling between the apo and holo structures of a protein via rejection sampling, which circumvents the issue that standard OT coupling will face due to lack of holo samples for some structures. This allows the model to better approximate the optimal flow and achieving SOTA performance on protein-ligand docking at <1A resolution. I recommend this work for acceptance.


[1] Corso, Gabriele, et al. "Diffdock: Diffusion steps, twists, and turns for molecular docking." *arXiv preprint arXiv:2210.01776* (2022).

[2] Lu, W., Zhang, J., Huang, W. *et al.* DynamicBind: predicting ligand-specific protein-ligand complex structure with a deep equivariant generative model. *Nat Commun* **15**, 1071 (2024). https://doi.org/10.1038/s41467-024-45461-2

[3] Corso, Gabriele, et al. "Deep Confident Steps to New Pockets: Strategies for Docking Generalization." *arXiv preprint arXiv:2402.18396* (2024).

---

### Official Review · Reviewer_iXqn · 2024-06-12
**Good paper with a novel method**

**Rating:** 7
**Confidence:** 2

**Review:**

In this work the authors propose a novel method, Unbalanced Flow Matching, that leverages diffusion models for flexible molecular docking of protein-ligand compounds. At variance with standard Flow Matching, this generative framework relaxes some assumption allowing to learn simple maps that transport the flow. The authors are able to provide bound estimates of the sample efficiency vs the approximation quality of the method. On benchmark protein structure datasets, the method shows increased performances against other approaches and sensibly faster runtimes.

In general the work is well written and novel. I would suggest the author to move (page limit permitting) section B in the main to make the manuscript more accessible.

---

### Official Review · Reviewer_7ctg · 2024-06-12

**Rating:** 8
**Confidence:** 3

**Review:**

This paper presents a method for incorporating unbalanced flow-matching to the task of flexible docking. The goal is to allow conformational flexibility in the protein (as opposed to rigid docking) while also leveraging known information about the distribution of apo structure (as opposed to starting from scratch as in cofolding). By relaxing the mass transfer constraint in flow-matching and rejecting undesirable samples from q(x_0), the authors formulate a more efficient flow than can then be reweighted.

Pros:
- Well written with clear motivation that highlights the impact and novelty compared to previous rigid-protein and cofolding techniques, as well as why traditional flow-matching is poorly suited for this task
- Paper is highly relevant given AlphaFold3’s recent results on cofolding protein-ligand complexes
- Extensive comparisons are shown against both traditional docking tools and generative approaches, showing notable improvement in protein RMSD and inference time while retaining comparable performance on ligand RMSD


Other Comments
- This is the first approach I’ve seen that combined (practically) independent diffusion and flow models, perhaps some more discussion could be warranted on how and why this should work?
- It is not clarified until the last page that q(x0) structures are computational predictions from ESMFold, I think specifying this fact earlier would help with clarity.
- It remains unclear to me if i) there are multiple structure predictions for a given protein and X structures are removed based on the the c_dock rmsd cutoff or ii) the cutoff is removing entire apo-holo pairs with large rmsds from the dataset. The first case makes more sense for this application, especially given the author’s discussion of the conformational diversity of apo states, but to my knowledge ESMFold cannot predict an ensemble of structures for a given sequence. In the latter case, some more detail is warranted in how this may reduce the effective size of the dataset as a function of c_dock
- There is substantial discussion of reweighting unbalanced flow-matching based on rejection sampling, however it’s not clear to me how this is incorporated into the loss or in the inference procedure
Is the flow-matching procure by itself deterministic (assuming you are starting from an unnoised ESMFold prediction) and diversity and in prediction structure coming from the diffusion on the ligand alone? In that case I would assume diversity of predicted protein structures to be fairly low.
- The work of Lu et al. 2024 contains a somewhat similar workflow based on flexible docking from initial Alphafold structures. This paper is distinct in several ways, but the work should be cited in the intro (currently only in the appendix) and a comparison would be interesting as well

Small point:
- Appendix should be submitted as a separate doc
- Small typo in confidence model section “of mthe anifold docking”